# Health Communication through Chinese Media on E-Cigarette: A Topic Modeling Approach

**DOI:** 10.3390/ijerph19137591

**Published:** 2022-06-21

**Authors:** Qian Liu, Yu Liang, Siyi Wang, Zhongguo Huang, Qing Wang, Miaoyutian Jia, Zihang Li, Wai-Kit Ming

**Affiliations:** 1School of Journalism and Communication, National Media Experimental Teaching Demonstration Center (Jinan University), Jinan University, No. 601, West Huangpu Avenue, Guangzhou 510632, China; tsusanliu@jnu.edu.cn (Q.L.); yuliang@jnu.edu.cn (Y.L.); secretbaser99@stu2020.jnu.edu.cn (Q.W.); jmyteden@stu2021.jnu.edu.cn (M.J.); 2002715lzh@stu2020.jnu.edu.cn (Z.L.); 2Department of Public Health and Preventive Medicine, School of Medicine, Jinan University, No. 601, West Huangpu Avenue, Guangzhou 510632, China; wangsiyi@stu2019.jnu.edu.cn (S.W.); hzg913290007@163.com (Z.H.); 3Department of Infectious Diseases and Public Health, Jockey Club College of Veterinary Medicine and Life Sciences, City University of Hong Kong, To Yuen Building, 31 To Yuen Street, Hong Kong, China

**Keywords:** health communication, e-cigarettes, news, China, LDA, vaper

## Abstract

Background: Electronic cigarettes (e-cigarettes) have been a newsworthy topic in China. E-cigarettes are receiving greater consumer attention due to the rise of the Chinese e-cigarettes industry. In the past decade, e-cigarettes have been widely debated across the media, particularly their identity and their health effects. Objective: this study aims to (1) find the key topics in e-cigarette news and (2) provide suggestions for future media strategies to improve health communication. Method: We collected Chinese e-cigarettes news from 1 November 2015 to 31 October 2020, in the Huike (WiseSearch) database, using “e-cigarettes” (Chinese: “电子烟”) as the keyword. We used the Jieba package in python to perform the data cleaning process and the Dirichlet allocation (LDA) topic modeling method to generate major themes of the health communication through news content. Main finding: through an analysis of 1584 news articles on e-cigarettes, this paper finds 26 topics covered with 4 themes as regulations and control (*n* = 475, 30%), minor protection (*n* = 436, 27.5%), industry activities (*n* = 404, 25.5%), and health effects (*n* = 269, 17%). The peak and decline of the number of news articles are affected by time and related regulations. Conclusion: the main themes of Chinese news content on e-cigarettes are regulations and control, and minor protection. Newspapers should shoulder the responsibilities and play an important role in health communication with balanced coverage.

## 1. Introduction

E-cigarettes are battery-powered nicotine delivery devices, which is an effective and harmless substitute of a smoking cessation program [1] and could aid in preventing smoking-related deaths [2,3,4]. However, there are many scientific studies to prove that e-cigarettes are harmful to human health and attract non-smokers. Moreover, e-cigarette can be a gateway to smoking and also contain ingredients that can cause addiction and respiratory symptoms, such as nicotine and other chemical hazardous agents or ultrafine particles [5]. Its aerosols have adverse effects on immune function [6,7,8] and the cardiovascular system [9,10,11]. It has been estimated that 16.9 million adults were using e-cigarettes, and the prevalence of use among young adults increased from 2.0% (2015–2016) to 2.7% (2018–2019) in China [12]. This finding highlights the critical relevance of public health communities and policymakers’ urgent efforts to formulate regulations and public education strategies for various demographics and provide essential evidence for upgrading Chinese smoke-free zone regulations in 2018. The state has issued a series of regulations to control the growing trend of vaping among minors and regulate the industry. State Administration for Market Regulation (SAMR, as listed in Appendix A, Table A1) and State Tobacco Monopoly Administration (STMA) applied a ban on producing and selling e-cigarettes to adolescents on 31 August 2018 [13]. Subsequently, they urged e-cigarette manufacturers, sellers, and e-commerce platform operators to terminate online sales channels, pull off e-cigarette products, and withdraw online advertisements on 1 November 2019 [14]. However, leading brands of most tobacco product types use social media extensively, which will attract young people to buy e-cigarettes [15]. Therefore, it is extremely important to explore the following two research questions: (1) what are the main topics of e-cigarette-related news report in China? (2) what are possible suggestions for media strategies to improve health communication related to e-cigarette?

As the primary source of information for the public, the function of media goes beyond disseminating information. News coverage is significant given the news media’s ability to shape public awareness and perceptions of the salience of products and policies [16,17], which may influence the public use intention [18,19]. Health-related news often contains conflicting information on health topics, including disease risk factors, disease prevention, screening, and treatment recommendations. In recent years, news coverage of new tobacco products, including e-cigarettes and smokeless tobacco, have made the use of these products induce both positive and negative trends. E-cigarettes are being promoted on social media as better tasting and cleaner, which may motivate experimenters to use e-cigarettes [20]. Moreover, research shows that social media use linked to increased e-cigarette use through online e-cigarette advertising exposure [21]. Some research shows that as one effective tobacco control measure, mass media, particularly news media, are designed to reduce tobacco use through agenda setting [22,23]. In the United States, content analysis reveals people’s great interest in e-cigarettes, especially the supervision of e-cigarettes and its common main topics on e-cigarettes were policy/regulatory issues of e-cigarettes, health effects, and e-cigarettes prevalence, and news stories more frequently mentioned passive themes (potential risks or concerns) than positive (benefits) themes [24,25]. Another study found that getting around smoke-free legislation, risk and uncertainty, healthier choice, celebrity use and price were five key themes in the United Kingdom and Scotland; furthermore, their themes and content were more positive than US news theme [26]. Numerous studies suggest that mass media can influence the spread of e-cigarettes [20,21]. Nevertheless, those studies have utilized content analysis, a research method that can analyze a limited sample. Another research method, the Latent Dirichlet Allocation (LDA), used in computational communication, can realize mass text analysis in a data-driven approach and extract frameworks and topics from a large amount of news texts. Besides, the prominence of e-cigarettes in the media has not been given enough attention [27]. Some relevant studies are listed in Appendix A, Table A2.

The main objectives of the study are to find the key topics in e-cigarette news and to provide suggestions for future strategies to improve health communication. Therefore, we first examined how e-cigarettes are reported in Chinese news and concluded the main themes by collecting a total of 1584 e-cigarettes news and analyze with the LDA topic modeling method, a generative approach for collections of discrete data [28]. Then, we offered recommendations on e-cigarette-related health communication strategies by comparing regulation and news volume changes in China and analyzing relevant e-cigarette policies in nations all over the world.

## 2. Methods

### 2.1. Data Collection

In order to gain insights about health communication through Chinese media platforms, we collected Chinese e-cigarettes news from 1 November 2015 to 31 October 2020. We chose the keyword “e-cigarettes” and searched the Huike (WiseSearch) database, one of the most professional leading Chinese media databases founded in 1998. This database collected over 96 million news items daily, over 1600 print media, and 50,000 internet media offer news to this database [29].

### 2.2. Model Introduction

The reason why we chose LDA topic modeling is because it is a type of topic modeling method that has been utilized in a number of studies for text mining [30], or in fields of psychology and medicine [31,32], as well as our team’s previous research on third-hand smoke [33]. This unsupervised machine learning technique automatically generates topics from documents and categorizes similar documents to one or more of these topics based on the distribution of words, so we can implement a lot of text analysis. The model uses the observed documents and words to infer the hidden topic structure, creating per-document topic distributions, P(topic|document), and per-topic word distributions, P(word|topic) [34]. The basic idea is that documents are represented as random mixtures over latent topics, where each topic is characterized by a distribution over words [28].

LDA assumes the following generative process for each document *w* in a corpus D:
Choose N ∼ Poisson (ξ);Choose θ ∼ Dir (α);For each of the *N* words $w_{n}$:
(a)Choose a topic $z_{n}$~Multinomial (*θ*);(b)Choose a word $w_{n}\;$ from $p(w_{n}\;|z_{n},\beta )$, a multinomial probability conditioned on the topic $z_{n}$.



α is a parameter of Dirichlet prior on the per-document topic distributions and β is a parameter of Dirichlet prior on the per-topic word distributions, θ is a k-dimensional Dirichlet random parameter. N is the word of the document where $w_{n}$ is the nth word in it. For a more detailed model explanation, please see the document by David M. Blei [28].

To better understand the news data on e-cigarettes, we use LDA topic modeling method to study the main topics of the news content. Our paper applied LDA to generate major themes of health communication through news content. There is a three-level hierarchical Bayesian model in the LDA method, and words are a random mixture of multiple latent topics. Different topics are related to a distribution over words [28]. By applying LDA modeling, we can calculate, with Gibbs Sampling, different topics from the text dataset [35].

### 2.3. Processing

The screenshot of the Huike database search page in Figure 1 to describe the searching of our data as follows:

A total of 1584 e-cigarettes news were collected from Huike. We cleaned and prepared the data for LDA topic modeling as illustrated in Figure 2 with Python 3.0. We use the Jieba package in python (a Python package used for Chinese language processing) to separate words, as Chinese words are connected [36,37]. We also removed the null and repeated information to conduct further study. Common Chinese stop words with less information such as “a,” “the,” or “very,” were excluded to generate meaningful result. Then, we applied the term frequency-inverse document frequency (TF-IDF) process to the data to weight the significance of a word in one specific article [38].

Choosing a proper topic number as a parameter for further analysis is significant for the LDA topic modeling approach in order to avoid having too many or too few topic numbers that are either difficult to interpret or unable to distinguish one another. Our study looked for the optimum number of topics for LDA by calculating the coherence score [39]. This measure compares the degree of semantic similarity for different topics by calculating every two context vectors’ average of cosine values [40]. We use Gensim (a Python package) to calculate a coherence value and find an optimal number of 26, as shown in Figure 3 [41]. Therefore, we chose 26 as the proper parameter for the topic number and set λ = 1 to conduct the analysis with LDAvis tool in Python. Each topic was named according to the top keywords, respectively, as shown in Table 1. We also concluded with four major themes to further analyze the major emphasis of health communication.

We visualize the topic allocation as circles in Figure 4 and Figure 5 to demonstrate the connections. Each circle represents one topic. The distance between circles were determined by topic similarities [28].

## 3. Results

To better understand the relationship between different topics, we visualize the 26 topics in Figure 4. This figure shows 26 circles that represent respective topics. We can obtain the overall prevalence by computing areas of all of the circles compared with the reference area below (the area is the prevalence). Their prevalence can also be seen from the statistical results on the right side of Table 1. Intertropical distances are represented by multidimensional scaling on a 2D plane [42]. Figure 4 answers two important research questions: (1) what is the meaning of each topic? (2) how prevalent is each topic? In the Figure 4, the circles 1, 2, 3, and 4 are relatively large, indicating that this topic accounts for a large proportion of all of the topics. There is overlap between circles 1 and 2 in the Figure 4, indicating that there is a relationship between the two topics. The interaction between different themes illustrates the correlation between the popular topics. The principal components PC1 and PC2 represent the transverse axis and longitudinal axis, respectively [43].

As shown theme 3: more harmful substances than combustible tobacco (5.1%), topic 10: addiction and harms (4.3%), topic 12: toxic vapor (4.2%), and topic 18: higher risk of illness influenced by vaping (3.4%) positioned close to each other, thus form theme 3: health effects. As illustrated in Figure 4, some circles are closer to others, which shows how topics are related to each other. Closer the circles are plotted, more related the topics are, because circles’ centers are calculated by computing the topics distance and projection of inter-topic distances onto two dimensions.

According to the visualization and interpretation, we grouped the topics into 4 major themes displayed in Table 1 for e-cigarettes news [43]. Keywords are listed on the right side of the table for clear display, along with the quantity and frequency of each topic name.

To further discuss each topic, we displayed the top 30 keywords as relevant terms, shown in Figure 5. For topic 1, they are mainly about Electronic, Industry, Market, and others. They had the highest proportion in their period; therefore, we present them as examples for demonstration. The overall term frequency and estimated frequency of term Electronic are highest and followed by Industry and Market. The word frequency distribution was relative to the entire corpus by the system. The blue bar presents the overall term frequency, and the red bar presents the estimated frequency of a specific topic. In topic 1, Chinese mass media preferred to talk about what treatment the patients can receive at the end of life. By this approach, illustrated in previous literature, we could interpret topics’ content [44].

Table 2 shows Chinese regulations concerning e-cigarettes recent years. Figure 6 indicates the time and the quantity of news coverage from 2018 to 2020. Prior to the emergence of relevant policies, the level of news coverage on e-cigarettes has been sluggish. The number of news articles peaks and wanes, affected by time and related regulations and inspections, as shown in Figure 6, mainly including local and national regulations. In the month that the regulation was released, the number of news articles on e-cigarettes rose to a high level.

Media sources facilitate us to build the communication map from the origin, where we list news agencies, released most e-cigarette news in Figure 7 and gives the evidence that China News Service covered the most reported news during 2015 to 2020, and then followed by Beijing Evening (Digital News) and Dajiang Evening. According to Figure 7, how different active media were in e-cigarette news can be observed. Shanghai Morning Post, Beijing News, and other news agencies were also active at the time of collection. Traditional news sources and online news sources both contribute to the health communication on e-cigarettes.

## 4. Discussion

In our study, we we discovered that e-cigarettes are becoming more prominent in the health communication media agenda. The result of the topic modeling assists us in comprehending e-cigarette news coverage in the Chinese press. To explore the overall themes of e-cigarette in health communication of media and proposals for media strategies to promote health communication, our study retrieved relevant news from online and print outlets from 2015 to 2020.

Our results revealed that the four major themes emerged: control and regulation, minor protection, industry (brand) development, health effect. Most domestic news covered the negative aspects of e-cigarettes. Accept for theme 1, which mainly involves the development of the industry, the other three themes, and the following topics, especially in topic 3 (*n* = 81, 5.1%), have constructed a relatively negative image of e-cigarettes: e-cigarettes are not conducive to human bodies, especially youth health. Therefore, the nation has issued regulations and taken actions to regulate the sales of e-cigarettes. Such an agenda may change readers’ awareness, perception, and even behavior toward e-cigarette use that people may refuse to vape [45,46].

One main theme of our study is regulations and control, accounting for 30% of all news. The number of new coverage fluctuates with the release of policies and activities (Figure 6). SMAR and STMA issued 2 youth-related policies separately on 28 August 2018 (Notice on Prohibiting the Sale of Electronic Cigarettes to Minors) and 1 November 2019 (Notice on Further Protecting Minors from Electronic Cigarettes), resulting in the number of news coverage reaching the peak. The theme of regulations and control appears most frequently, replicating prior research findings [47]. Those contents involved the nationwide regulations and special actions to regulate the e-cigarettes market.

Another prominent theme from our result is minor protection (taken up 27.5%), consistent with the intent of the WHO’s advocacy and SMAR and STMA’s e-cigarettes deeming regulations. Considering that May 31 is World No Tobacco Day, and its theme in 2020 is “preventing youth from industry manipulation and preventing them from tobacco and nicotine use [48], news from May to June frequently discussed issues related to youth and e-cigarettes. Since adolescent vaping may represent a risk factor for subsequent cigarette smoking, traditional media must cooperate with regulatory measures to the public, particularly strengthening the guidance for youth [49,50,51].

Regulations and special actions impact the theme of industry (brand) activities which accounted for 25.5% news content (shown in Table 1). Despite the fact that the top 30 most relevant terms have a strong correlation with market and industry development (as shown in Figure 5), because companies were not allowed to advertise e-cigarettes online, topics focused on current domestic market development and financing of the e-cigarettes industry and corporate brands under the control of policies, rather than promotion for e-cigarettes products. Asymmetry in health communication could be caused by skewed coverage of monopolistic e-cigarette brands or one-sided coverage of the benefits and drawbacks of e-cigarettes. Traditional media shoulders a vital role in the public agenda of e-cigarettes. Health communication should avoid the influence of capital and industrial involvement.

In terms of health effects (17% in total), news media functioned as public health education and an important pathway for preventing smoking [52], informing the potential impact of vaping. Those contents quote researchers and institutions in professional fields to increase the credibility of health information [53]. Almost all news stories emphasize the harmful effects of vaping on human health in China, despite the uncertainty disclosed by scientists and the discussion of both benefits and risks over media were observed in the past US and the UK research [24,25,26]. Outnumbered negative coverage may confuse conventional smokers who attempt to quit smoking through the means of e-cigarettes. Thus, framing a balanced coverage containing multiple reporting and a mix of viewpoints is also a new challenge for Chinese media. News media need to strengthen guidance and respect scientific research facts on e-cigarettes.

The prevalence of news stories emphasizing the dangers of e-cigarettes may raise public awareness of the dangers of e-cigarettes. The lack of information about the dangers of vaping and the prevalence of young e-cigarette addiction may also contribute to e-cigarette addiction, particularly among adolescents, so news organizations must have the responsibility of providing balanced coverage of health issues.

Irrespective from the findings of the news themes, we also noticed that the agenda remains salient. The official news agencies covered and released those e-cigarettes news (Figure 6 and Figure 7), shifting the public attention to e-cigarettes. Moreover, news agencies have digital news platforms or social media accounts, releasing e-cigarette news through these channels. Although some scholars have noticed limitations of newspaper and their influences in China and stated that e-cigarettes have not grown into the top topic for media agenda [27], however, the fact is that the agenda will flow from traditional media to digital media, and online discussion for e-cigarettes will be formed in the new media era. News media continue to guiding public opinions at present; thus, they should continue to play the role of a medium to inform and persuade public.

Government policymakers, the media, and the general public formed a circle in this process, and one can influence the other. Media serve as a conduit for government information, guiding public debate to some extent and providing politicians with insight into public problems.

The news media, which serves as the government’s megaphone, is responsible for e-cigarette health communication, particularly in terms of clarifying associated policies around the world. Different countries have diverse attitudes toward e-cigarettes, and the news content that is disseminated to the public differs as well. Countries such as the United Kingdom are supportive of e-cigarettes, press reports about them frequently focus on the benefits of e-cigarettes and the nation’s support [54,55]. More other countries, such as the USA, Singapore, South Korea, Thailand, India, German, Australia, and Canada, have a strictly controlled attitude toward e-cigarettes, releasing more negative news about e-cigarettes [56,57,58,59,60,61,62,63,64,65,66,67]. These news stories subtly impact popular perceptions of e-cigarettes in the long run.

In addition, e-cigarette coverage either transmits voices from the government or delivers the voices of other stakeholders. E-cigarette news often covered the attitudes and behaviors of experts, smokers or vapers, youth or parents, e-cigarette practitioners, and other stakeholders on e-cigarettes. Media coverage enhances the transfer of information and facilitates stakeholders better comprehend each other’s perspectives.

## 5. Limitation

To ensure research generalizability, our study uses a large sample size in the recent five years. Nevertheless, several limitations warrant a mention. Owing to the fact that Huike (WiseSearch) database only contains text and LDA fails to analyze too short articles, news contents from mass media through new media platforms employing images, and short text, such as microblogs, may be missed. Furthermore, owing to topic model analysis extracts key topics from news texts presenting as keywords, important statements with few coverages could be lost in the analysis process. Based on this coverage’ analysis, we cannot fully understand the extent to which the news coverage affects audiences in China. In this regard, audience-centric studies are necessary. Moreover, there is a tendency for people acquire e-cigarettes information from social media or web forums, future research should study how frames of news stories converted in multiple media channels.

## 6. Conclusions

This paper intends to explore news coverage concerns for e-cigarettes in China and generated four main themes by LDA analysis, which are regulations and control, minor protection, industry activities, health effects. Based on a comparison of the number of news and related policy changes in China, and further analysis of related e-cigarette policies in countries around the world, we propose the following media strategies suggestions:

(1) Skewed coverage of monopolistic e-cigarette brands or one-sided coverage of the pros and cons of e-cigarettes could create an imbalance in health communication, and how to maintain a balanced coverage requires multiple reporting and a combination of opinions. Overwhelming news articles underscore the potential harm to e-cigarette may raise public awareness of the risk of e-cigarettes. The lack of reports on the dangers of vaping and youth e-cigarette addiction may also lead to e-cigarette addiction, especially for adolescents. Therefore, newspapers should shoulder responsibilities and play important roles in health communication with balanced coverage.

(2) Newspapers should shoulder the responsibilities of e-cigarette health communication, especially the explanation of changes in important tobacco rules. Newspapers need to play an important role in educating citizens about health literacy and publicizing industry norms.

## Figures and Tables

**Figure 1 ijerph-19-07591-f001:**
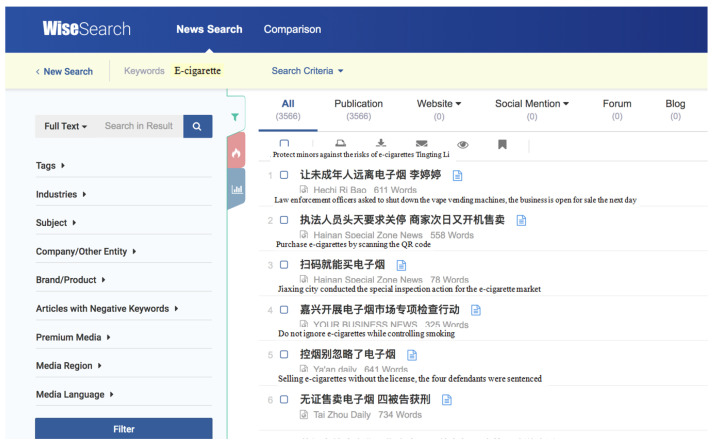
The screenshot of Huike database search page.

**Figure 2 ijerph-19-07591-f002:**
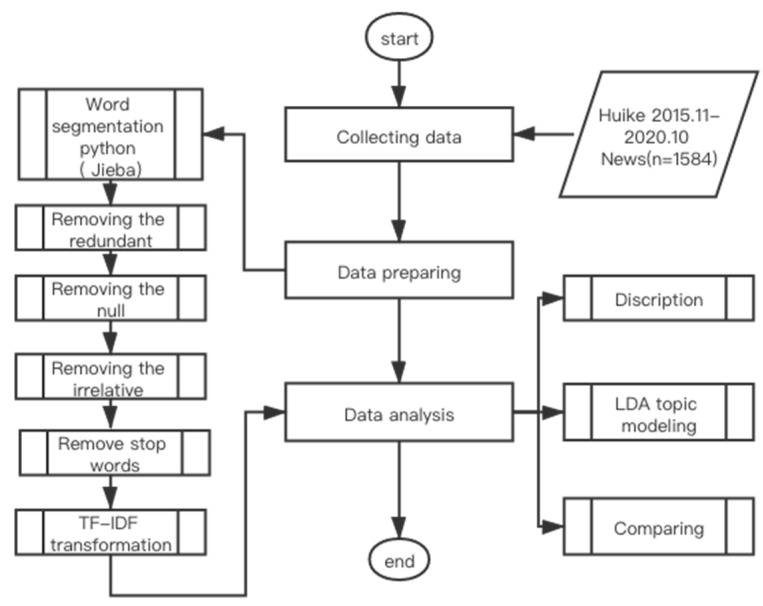
Data processing chart.

**Figure 3 ijerph-19-07591-f003:**
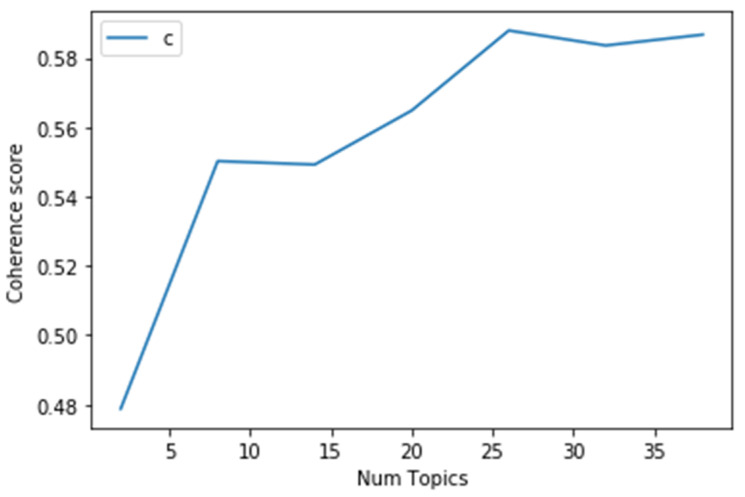
Coherence score for different topic number.

**Figure 4 ijerph-19-07591-f004:**
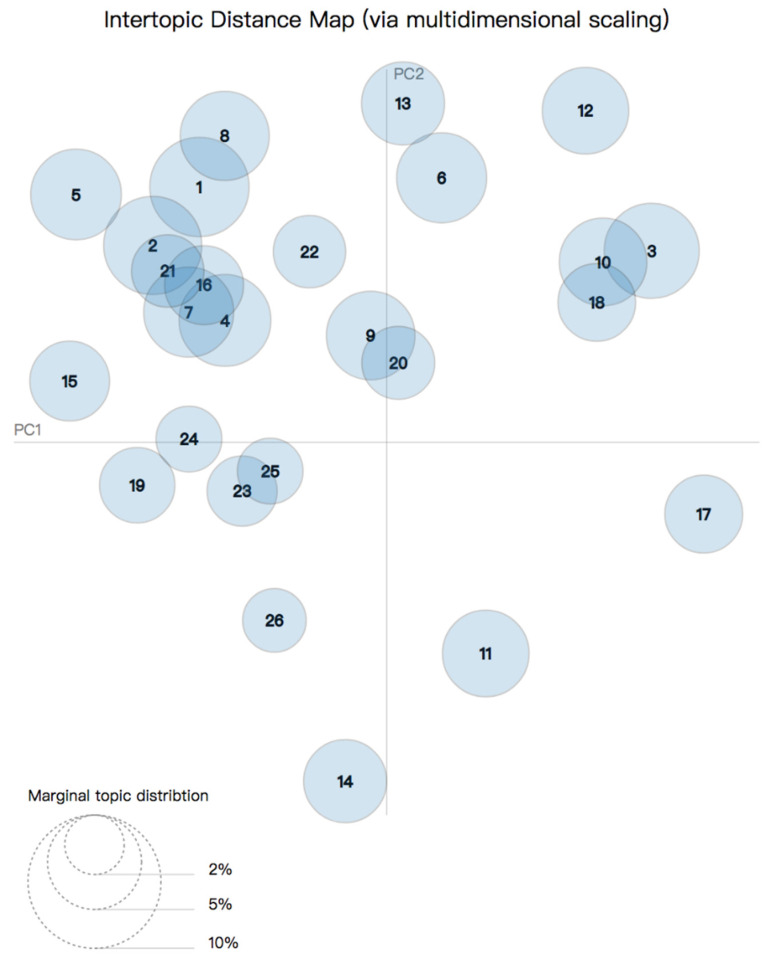
Inter-topic distance map for e-cigarettes (Topic numbers are marked in the circle).

**Figure 5 ijerph-19-07591-f005:**
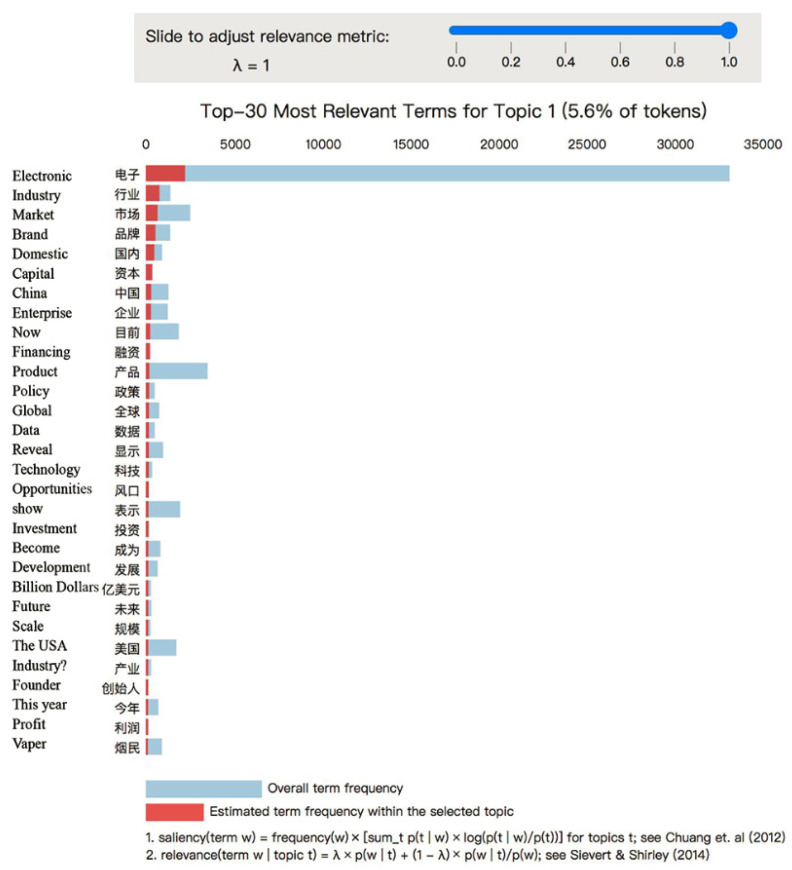
Top 30 most relevant terms for topic 1 (5.6% of Tokens) [42,43].

**Figure 6 ijerph-19-07591-f006:**
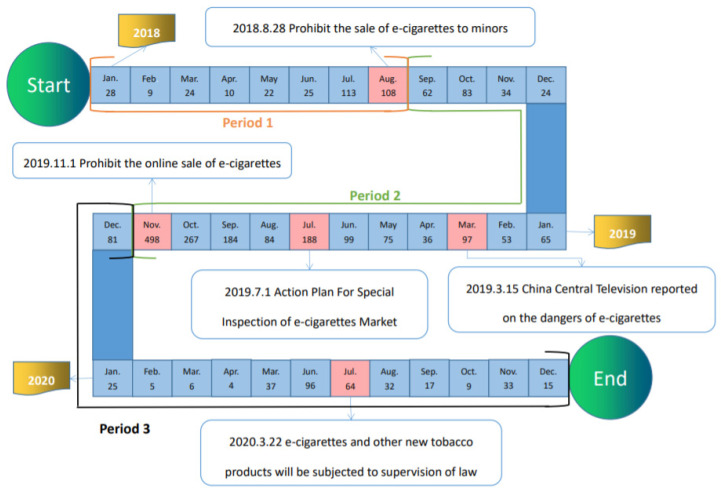
Timeseries of News Streams with Corresponding Policies during 2018–2020.

**Figure 7 ijerph-19-07591-f007:**
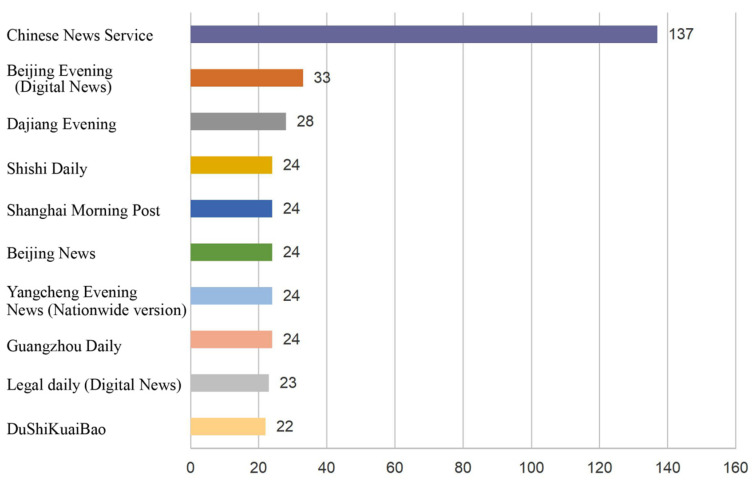
The most represented media sources for the collected news reports (N = 2886).

**Table 1 ijerph-19-07591-t001:** Topic classification and key words.

Theme, Topics, and Keywords	News Reports (N = 1584), *n* (%)
Theme 1: Industry activities	25.5%
Topic 1 The domestic market and financing situation of the industry and enterprises	5.6%
Keywords: e-cigarettes, industry, market, brand, domestic, capital, China, enterprise, current, financing	
Topic 7 Chinese company (Mike Weir) and products in the global market	4.6%
Keywords: company, e-cigarettes, market, tobacco, Mike, Weir, billion, global, international, increase	
Topic 8 The situation of practitioners selling e-cigarettes	4.5%
Keywords: e-cigarettes, reporter, purchase, business, indicate, a, find, emporium, use	
Topic 9 Accidents	4.4%
Keywords: the USA, e-cigarettes, use, case, product, illness, report, declare, report, ban	
Topic 16 Brand (Yueke) development	3.4%
Keywords: brand, a, trade, e-cigarettes, produce, product, already, company, Yueke, factory	
Topic 21 Industrial standard	3%
Keywords: e-cigarettes, Shenzhen, enterprise, China, industry, standard, development, product, global, technology	
Theme 2: Minor protection	27.5%
Topic 2 Regulations	5.4%
Keywords: e-cigarettes, minor, sales, notice, release, Internet, market, product, protection, country	
Topic 6 Inadequate supervision	4.6%
Keywords: e-cigarettes, health, tradition, harm, regulation, publicity, young people, existence, become, smoker	
Topic 11 Prevalence of youth e-cigarettes use	4.2%
Keywords: teenager, tobacco, smoke, tobacco control, health, China, crowd, ratio, research, use	
Topic 13 Sold in flavors that are attractive to youth	3.9%
Keywords: e-cigarettes, tobacco tar, smoke, cigarette, young people, flavor, present, feel, smog, friend	
Topic 17 Harms to minors	3.4%
Keywords: nicotine, illness, smoke, juveniles, cause, give rise to, influence, emerge, arise, health	
Topic 20 Regulations around the world	3%
Keywords: e-cigarettes, product, tobacco, intake, prohibit, Hong Kong, teenager, indicate, health, point out	
Topic 22 Sale near schools	3%
Keywords: e-cigarettes, student, kid, reporter, school, fid, parent, this kind of, nearby	
Theme 3: Health effects	17%
Topic 3: More harmful substances than combustible tobacco	5.1%
Keywords: e-cigarettes, nicotine, produce, content, harm, tobacco, tradition, contain	
Topic 10: Addiction and harms	4.3%
Keywords: quit smoking, e-cigarettes, assistance, nicotine, smoker, cigarette, smoke, smoking addiction, health, indicate	
Topic 12: Toxic vapor	4.2%
Keywords: electronic, nicotine, cigarette, health, intake, contain, product, a sort of, atomize	
Topic 18: Higher risk of illness influenced by vaping	3.4%
Keywords: research, e-cigarettes, find, risk, personal, use, tradition, a, evidence	
Theme 4: Regulations and control	30%
Topic 4: Enact legislation	4.8%
Keywords: e-cigarettes, supervision, country, tobacco, current, conduct	
Topic 5: Prohibit online sales and remove related products	4.6%
Keywords: e-cigarettes, platform, sales, e-commerce, product, reporter, channel, undercarriage, related, offline, release, brand, represent	
Topic 14: Prevention and control public policy	3.9%
Keywords: vape, regulation, tobacco control, prohibit vaping, range, bring into, prohibit, place, public place, control	
Topic 15: Prevention and control actions	3.6%
Keywords: e-cigarettes, juveniles, market, sell, inspect, carry out, unite, overall, earmarked	
Topic 19: Illegal sales of e-cigarettes	3.2%
Keywords: sell, pod, cigar, tobacco monopoly, tobacco, illegal operation, late-model, products, burn, e-cigarettes	
Topic 23: Control of quitting vaping in public places	2.8%
Keywords: e-cigarette, prohibit, stipulate, Beijing City, manage, relevant, explicit, behavior, public place, conduct	
Topic 24: Illegal smuggling of marijuana drugs in e-liquid	2.4%
Keywords: custom, bar, smuggling, police, electronic, find, marijuana, seized, suspects, drug	
Topic 25: Penalties for e-cigarettes companies and warning labels requirement	2.4%
Keywords: e-cigarettes, logo, vape control, Shenzhen, vape, nationwide, ticket, product, stipulate	
Topic 26: Illegal use of e-cigarettes	2.3%
Keywords: safety, airplane, passenger, flight, survey, incident, aircrew, airport, cabin, Air China	

**Table 2 ijerph-19-07591-t002:** Chinese regulations about e-cigarettes.

Date	Regulations
1 June 2015	Regulations of Beijing Municipality on smoking control
28 August 2018	An Announcement on Prohibiting the Sale of e-cigarettes to Minors
1 November 2018	Administrative measures of Xi’an city on smoking control
1 January 2019	Regulations of Hangzhou Municipality on smoking control in public places
26 June 2019	Regulations of Shenzhen Special Economic Zone on smoking control
22 July 2019	A plan to regulate e-cigarettes through legislation produced by National Health Commission
26 September 2019	Regulations of Wuhan Municipality on smoking control
1 November 2019	An Announcement on Further Protecting Minors from e-cigarettes
31 December 2019	Notice of Shanghai market supervision and Administration Bureau on defining the specific scope of outdoor advertisements that may have adverse effects in this city
22 March 2021	The implementation regulations of the tobacco monopoly law will be amended: new tobacco products such as e-cigarettes will be included in the supervision
26 November 2021	State Council amend the tobacco monopoly law of the People’s Republic of China State Council amend the tobacco monopoly law of the People’s Republic of China
1 December 2021	State Administration for Market Regulation and Standardization Administration of the People’s Republic of China published National Standard for E-Cigarettes.
2 December 2021	State Tobacco Monopoly Publicly Solicit Opinions on the “Administrative Measures on E-Cigarettes (the Exposure Draft)”
1 May 2022	State Tobacco Monopoly implement “Administrative Measures on E-Cigarettes”

## Data Availability

Data available upon requirement.

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
