# Peer review of "Health Communication through Chinese Media on E-Cigarette: A Topic Modeling Approach"

_ijerph, 2022, doi:10.3390/ijerph19137591_

Round 1

Reviewer 1 Report

1. Abstract needs to modify be more quantitative. You can absorb readers' consideration by having some numerical results in this section. Moreover, numerical results should be presented in the conclusion.
2. Introduction section needs to be re-written to improve its quality and readability. My suggestion is to divide the introduction into three subsections: motivation and incitement, literature review and contribution and paper organization.
3. It helps to appreciate the paper by having a related review section. The authors should consider more recent research done in the field of their study. Please address the literature systematically. If possible, the authors can give a table pinpointing the advantage or limitations of each work.
4. The article missed presenting the research novelty. The authors should provide enough proof to convince the reader of superiority of the proposed schemes over the existing works.
5. There is no discussion of user requirements, technological options and support for the decisions made at the design (LDA). The authors should include more technical details and explanations.
6. The conclusion should be rewritten to clarify your contribution. Please points out some insufficiency and limitation that needs further improvements in the conclusion. Moreover, formats of reference list lack consistency.
7. The authors did not provide solid achievements in this manuscript since this paper seems to be a somewhat incremental piece of work based on earlier research results [A].

[A] Q. Liu, Q. Chen, J. Shen, H. Wu, Y. Sun and W. Ming, Data Analysis and Visualization of Newspaper Articles on Thirdhand Smoke: A Topic Modeling Approach. JMIR Medical Informatics, vol. 7, no. 1, 2019.

Reviewer 2 Report

The contribution would benefit from a proofreading and a better articulation of the methodological framework. The rigour of the topic is obscured by a lack of explanation of the algorithm in use in the Phyton package referred to, and while this may be clear to an expert audience it is not clear to everyone else. 

Reviewer 3 Report

The paper can be accepted after major revisions. Please see the attached file

Reviewer 4 Report

This article collects and organizes news about e-cigarettes in China, and analyzes these news reports to understand the meaning of e-cigarettes in Chinese news reports. This topic is useful for studying whether news reports can improve public awareness of the risks of e-cigarettes. It has certain significance and has certain academic research value. However, there are still some deficiencies or problems that need to be improved in this paper. The details are as follows:

1. The paper proposes LDA topic modeling when analyzing news data about e-cigarettes, but the paper does not explain the reasons for choosing this method.

2. The analysis in Figure 3 is slightly insufficient in the text. Please increase the analysis of the content and data in Figure 3.

3. As stated in the text, the LDA topic modeling method is not suitable for short texts, so please add a description of the impact on the conclusions in the text in the absence of data analysis of short news texts.

4. There are some minor errors in English expressions in the text. For example, in the result of the third part, "To better understand the relationship of different topics" should be changed to "To better understand the relationship between different topics", "We can get overall prevalence by computing areas of" should be changed to "We can get the overall prevalence by computing areas of", etc.

5. Please add future research directions in the conclusion section of the text.

Round 2

Reviewer 1 Report

This paper has edited and revised according to the reviewer's suggestions.

Reviewer 3 Report

The revised paper version is good. the paper can be accepted for publication